# Patient-derived frontotemporal lobar degeneration brain extracts induce formation and spreading of TDP-43 pathology in vivo

Sílvia Porta [1], Yan Xu[1], Clark R. Restrepo[1], Linda K. Kwong[1], Bin Zhang[1], Hannah J. Brown[1], Edward B. Lee [2], John Q. Trojanowski[1] & Virginia M.-Y. Lee[1]

The stereotypical distribution of TAR DNA-binding 43 protein (TDP-43) aggregates in frontotemporal lobar degeneration (FTLD-TDP) suggests that pathological TDP-43 spreads throughout the brain via cell-to-cell transmission and correlates with disease progression, but no in vivo experimental data support this hypothesis. We first develop a doxycycline-inducible cell line expressing GFP-tagged cytoplasmic TDP-43 protein (iGFP-NLSm) as a cell-based system to screen and identify seeding activity of human brain-derived pathological TDP-43 isolated from sporadic FTLD-TDP and familial cases with *Granulin* (FTLD-TDP-GRN) or *C9orf72* repeat expansion mutations (FTLD-TDP-C9+). We demonstrate that intracerebral injections of biologically active pathogenic FTLD-TDP seeds into transgenic mice expressing cytoplasmic human TDP-43 (lines CamKIIa-hTDP-43$_{NLSm}$, rNLS8, and CamKIIa-208) and non-transgenic mice led to the induction of de-novo TDP-43 pathology. Moreover, TDP-43 pathology progressively spreads throughout the brain in a time-dependent manner via the neuroanatomic connectome. Our study suggests that the progression of FTLD-TDP reflects the templated cell-to-cell transneuronal spread of pathological TDP-43.

[1] Center for Neurodegenerative Disease Research (CNDR), Institute on Aging, Department of Pathology and Laboratory Medicine, Perelman School of Medicine, University of Pennsylvania, Philadelphia, PA 19104, USA. [2] Translational Neuropathology Research Laboratory, Department of Pathology and Laboratory Medicine, Perelman School of Medicine, University of Pennsylvania, Philadelphia, PA 19104, USA. Correspondence and requests for materials should be addressed to V.M.-Y.L. (email: vmylee@upenn.edu)

Deposition of TAR DNA-binding 43 protein (TDP-43), especially phosphorylated TDP-43 (pTDP-43), into cytoplasmic and intranuclear inclusions is the pathological hallmark of frontotemporal lobar degeneration (FTLD-TDP) and amyotrophic lateral sclerosis (ALS), as well as a common comorbid pathology in other neurodegenerative diseases, such as Alzheimer's disease (AD)[1–3]. The stereotypical distribution of TDP-43 pathology as disease progresses suggests a hierarchical regional spreading of TDP-43 aggregates over time[4–6]. The cell-to-cell templated propagation of Aβ, tau, and α-synuclein (α-syn) aggregates offers a common pathophysiological hypothesis for disease progression in TDP-43 proteinopathies and other neurodegenerative diseases, including AD, Parkinson's disease, and related synucleinopathies[7–12]. Although the mechanisms underlying progression in these diseases are unknown, increasing evidence supports a model where misfolded proteins released from a cell harboring pathological inclusions act on recipient cells to form de novo pathology by corrupting endogenous normal proteins to adopt pathological conformations. The reiteration of this process leads to cell-to-cell propagation of pathological proteins throughout the brain[13,14].

Cell-to-cell transmission of TDP-43 aggregates occurs in cell culture, and exosomes have been proposed to contribute to pathological TDP-43 propagation[15–17]. Furthermore, anterograde and retrograde transport of TDP-43 oligomers in cell culture provide evidence for neuron-to-neuron transmission of pathological TDP-43[15]. Currently, however, no evidence exists to demonstrate TDP-43 spreading in animal models. Here we show that pathological TDP-43 derived from FTLD-TDP brains induces formation of de novo TDP-43 pathology with subsequent spreading throughout the central nervous system in a regional- and time-dependent manner in experimental animal models.

## Results

**Enhanced TDP-43 seeding by cytoplasmic mislocalization.** To investigate intracellular seeding of TDP-43 with FTLD-TDP brain-derived pathological TDP-43, we established a cell-based system by generating doxycycline (Dox)-inducible QBI-293 stable cell lines expressing either green fluorescent protein (GFP)-tagged wild-type TDP-43 (iGFP-WT) or mislocalized cytoplasmic TDP-43 (iGFP-NLSm) due to inclusion of mutations within the bipartite nuclear localization signal[18]. Both GFP fluorescence and immunofluorescence (IF) using an anti-TDP43 antibody (Fig. 1a) showed transgene expression, which was suppressed in the absence of Dox (Dox−), whereas Dox treatment (Dox+) for 72 h resulted in the expression of either nuclear GFP-WT (Fig. 1a, upper panels) or cytoplasmic GFP-NLSm proteins (Fig. 1a, bottom panels). Protein extracts from subclones of iGFP-WT and iGFP-NLSm cell lines were obtained by sequential extraction with RIPA followed by urea buffer and analyzed by immunoblot. Subclones with comparable GFP-WT and GFP-NLSm expression levels were selected for further studies (Supplementary Fig. 1a). Densitometric quantification of TDP-43 immunoblots showed ~2.5-fold increase in the expression of GFP-WT and GFP-NLSm protein over endogenous TDP-43 in RIPA fractions (Supplementary Fig. 1b). Moreover, as described previously[18,19], overexpression of GFP-WT and GFP-NLSm proteins reduced endogenous TDP-43 levels by ~50 and ~40%, respectively (Supplementary Fig. 1b).

Using iGFP-WT and iGFP-NLSm cell lines, we evaluated whether TDP-43 intracellular aggregates can be induced when treated with sarkosyl-insoluble extracts containing pathological TDP-43 from FTLD-TDP postmortem frontal cortex (Supplementary Fig. 2). Sarkosyl-insoluble FTLD-TDP extracts or age-matched non-pathological brain extracts were analyzed by

immunoblot using an antibody specific for phosphorylation at Ser409/Ser410 and a C-terminus anti-TDP-43 antibody (Fig. 1b, left panel). FTLD-TDP brain extracts contained pathological TDP-43 protein including phosphorylated full-length TDP-43 and C-terminal fragments (CTFs) (Fig. 1b, left panel, lane #2) whereas no phospho-immunoreactive bands were detected in non-pathological brain extracts (Fig. 1b, left panel, lane #1). iGFP-WT and iGFP-NLSm cells were transduced with FTLD-TDP and non-pathological brain-derived extracts and the formation of TDP-43 aggregates was analyzed 3 days post-transduction (dpt). Immunoblot analysis of the sarkosyl-insoluble fraction from iGFP-WT and iGFP-NLSm cells revealed the accumulation of phosphorylated GFP-WT or GFP-NLSm proteins when cells were transduced with FTLD-TDP seeds (Fig. 1b, right panel). No pTDP-43 or CTFs were detected upon transduction with non-pathological extracts (Fig. 1b, right panel). iGFP-NLSm cells exhibited increased phosphorylated full-length GFP-NLSm protein and CTFs when compared with iGFP-WT cells (Fig. 1b). Despite the downregulation of endogenous TDP-43 protein levels in GFP-WT and GFP-NLSm overexpressing cells, a faint phospho-positive band was visible at ~43 KDa indicating that endogenous TDP-43 may also be recruited into the aggregates (Fig. 1b, arrow).

The formation of cytoplasmic TDP-43 aggregates in iGFP-WT and iGFP-NLSm cells transduced with sarkosyl-insoluble FTLD-TDP extracts was evaluated by IF at 3 dpt (Fig. 1c). While p409-410-positive aggregates were detected in both iGFP-WT and iGFP-NLSm transduced cells (Fig. 1c), quantification of the percentage of area occupied by p409-410 immunostaining showed a significantly higher burden of TDP-43 aggregates in iGFP-NLSm cells compared with iGFP-WT cells (Fig. 1d). These results indicate that cytoplasmic mislocalization of TDP-43 favors the process of seeding and formation of de novo TDP-43 aggregates.

**TDP-43 biochemical profile and seeding activity.** We used the iGFP-NLSm cellular model to screen for the seeding efficiency of insoluble brain extracts from sporadic FTLD-TDP and familial FTLD-TDP cases with mutations in either *Granulin* (FTLD-TDP-GRN) or *C9orf72* repeat expansion mutations (FTLD-TDP-C9+) genes (Supplementary Table 1) using an established protocol (Supplementary Fig. 2)[2]. iGFP-NLSm cells were transduced with equivalent amounts of insoluble TDP-43 measured using a C-terminal TDP-43 sandwich enzyme-linked immunosorbent assay (ELISA) (Supplementary Table 2). In general, higher levels of sarkosyl-insoluble TDP-43 from FTLD-TDP brains were detected by this TDP-43 ELISA since it detects both full-length and CTFs, whereas an N-terminus sandwich ELISA only recognizes full-length TDP-43[3,18]. Sarkosyl-insoluble extracts from non-pathological cases were used as a negative control. iGFP-NLSm cells were immunostained for p409-410 at 3 dpt, and the percentage of area occupied by p409-410 immunostaining was quantified as a measure of seeding activity. Pathological TDP-43 from both sporadic and familial FTLD-TDP cases showed higher seeding activity than controls with exception of two sporadic FTLD-TDP cases (Fig. 1e). Moreover, extracts from FTLD-TDP-GRN cases showed significantly higher seeding activity than sporadic FTLD-TDP cases (Fig. 1e), while there were no significant differences between FTLD-TDP-C9+ and sporadic FTLD-TDP or FTLD-TDP-GRN. Immunodepletion of TDP-43 from FTLD-TDP extracts significantly reduced seeding activity (Supplementary Fig. 3), indicating that pathological TDP-43 in the extracts is the likely source of seeding activity, although we cannot conclusively conclude that all the seeding activity is due entirely to pathological TDP-43. Despite the formation of

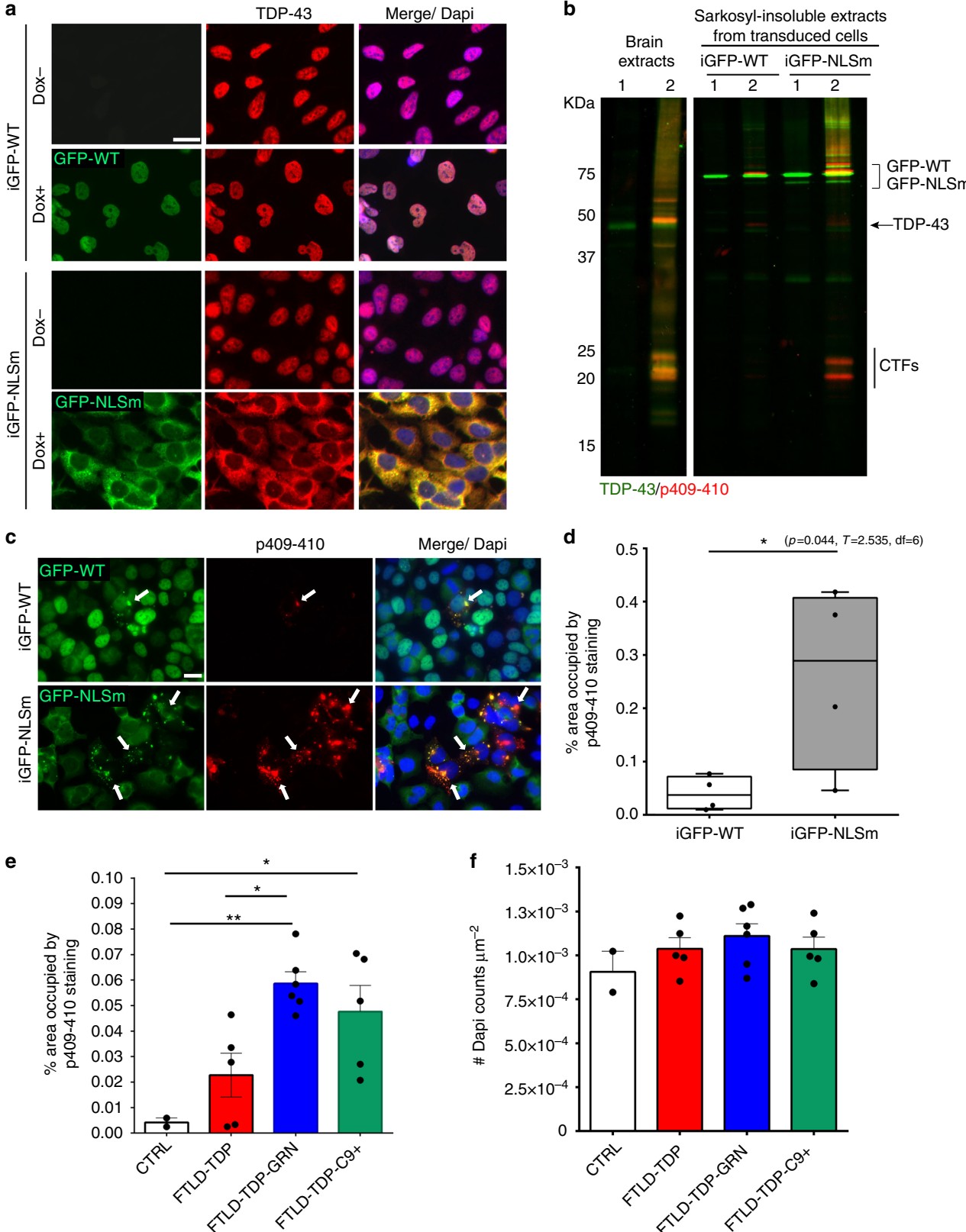

TDP-43 aggregates, no significant differences in cell viability were found at 3 dpt comparing iGFP-NLSm cells transduced with FTLD-TDP or CTRL extracts (Fig. 1f).

To investigate the biochemical basis of different seeding activities, we compared the TDP-43 banding pattern of sarkosyl-insoluble fractions by immunoblot (Supplementary Fig. 4a). There were no major differences between sarkosyl-insoluble brain extracts from sporadic or familial FTLD-TDP cohorts. As described in previous studies[20,21], the immunoblot analysis showed the presence of full-length TDP-43-positive

**Fig. 1** Inducible cytoplasmic TDP-43 cell-based assay for screening of FTLD-TDP brain-extract seeding activity. **a** Representative photomicrographs of IF analysis to detect TDP-43 proteins (red and merge) in iGFP-WT and iGFP-NLSm cells in the absence of Dox (Dox−) or after 72 h of Dox treatment (Dox+). Expression of GFP-WT or GFP-NLSm proteins (green) is detectable after Dox treatment (Dox+). Cells were counterstained with DAPI to label the nuclei. Scale bar = 20 μm. **b** Immunoblot analysis of the sarkosyl-insoluble fraction from CTRL and FTLD-TDP brains used as a seeds (left panel, lanes #1 and #2, respectively) and sarkosyl-insoluble extracts from iGFP-WT and iGFP-NLSm cells (right panels) transduced with CTRL or FTLD-TDP seeds at 3 dpt. A C-terminal TDP-43 antibody (C1039, green) was used to detect total TDP-43 protein and the phospho-specific Ser409/Ser410 mAb (p409-410, red) was used for detection of the insoluble/pathological TDP-43. GFP-WT or GFP-NLSm were detected at ~69 KDa, CTFs ~25–20 KDa, and endogenous TDP-43 at ~43 KDa (arrow). **c** Representative photomicrographs of IF analysis using the p409–410 (red and merge) in iGFP-WT and iGFP-NLSm cells (green and merge) transduced with FTLD-TDP extracts at 3 dpt. Arrows point to p409–410-positive aggregates. Cells were counterstained with DAPI to label the nuclei. Scale bar = 20 μm. **d** Plots show the percentage of area occupied by p409–410 immunostaining in iGFP-WT and iGFP-NLSm cells at 3 dpt. Box-and-whisker plots show the median (solid line) and whiskers indicate the minimum and maximum values, with individual points representing independent experiments ($n$ = 4) overlaid as black circles. Unpaired two-tailed Student's $t$ test was used for the analysis. *$P < 0.05$. In **e**, plots show the quantification of the percentage of area occupied by p409-410 staining in iGFP-NLSm cells transduced with sporadic FTLD-TDP (red), FTLD-TDP-GRN (blue), FTLD-TDP-C9+ (green), and CTRL (white) extracts and the number of DAPI counts μm$^{-2}$ in **f**. In **e**, **f**, bar plots show mean and whiskers s.e.m., with individual points representing a different brain extract overlaid as black circles. A one-way ANOVA followed by a Tukey's multiple comparisons test was used for the analysis; in **e**, treatment; $P = 0.004$, $F = 7.08$, degrees of freedom (DF) = 3. *$P < 0.05$ and **$P < 0.01$

bands at ~43 KDa and the CTFs with a pattern of three major bands between ~23 and 26 KDa and two minor bands at ~18–19 KDa (Supplementary Fig. 4a). However, immunoblots revealed a distinct CTF banding pattern in cases #4 and #5, corresponding to the two sporadic FTLD-TDP cases with lower seeding activity, where immunobands at ~18–19 KDa were missing (Supplementary Fig. 4a, b).

To elucidate whether differences in seeding activity correlates with the presence of distinct structural or conformational TDP-43 species, sarkosyl-insoluble "active" and "inactive" FTLD-TDP extracts were digested with increasing concentrations of proteinase K (PK)[16] and the resulting TDP-43 fragments were analyzed by immunoblotting (Supplementary Fig. 4c). Immunoblot analysis showed that full-length TDP-43 was easily degraded in "active" and "inactive" extracts, whereas CTFs, in particular the two lower Mr fragments, were relatively PK resistant (Supplementary Fig. 4c, d). Moreover, PK digestion led to the presence of a new TDP-43 immunoreactive band at <20 KDa in "active" but not in "inactive" sarkosyl-insoluble brain extracts (Supplementary Fig. 4c, d (band 6)). We also treated sarkosyl-insoluble extracts with increasing concentrations of GuHCl[22], which revealed the distinct banding pattern that distinguished "active" vs. "inactive" extracts, including the presence of intermediate TDP-43 products between ~37 and 25 KDa in the sarkosyl-insoluble fraction of "inactive" case (Supplementary Fig. 4e). Thus, among the FTLD-TDP cases we studied here, distinct immunobands of insoluble TDP-43 protein correlate with a distinct seeding activity in vitro, suggesting that different TDP-43 species or strains from FTLD-TDP brains may have different seeding capabilities.

**FTLD-TDP brain extracts seed cytoplasmic TDP-43 in vivo.** Several neurodegenerative disease proteins including Aβ, tau, α-syn, mutant huntingtin, mutant SOD1, and TDP-43 misfold and self-propagate through templated recruitment[23]. With respect to pathological TDP-43 within FTLD-TDP and ALS tissue extracts, propagation has been demonstrated only in cell culture models[15,16,24]. Thus we tested the hypothesis that human brain-derived pathological TDP-43 could induce the templated propagation and spreading of TDP-43 pathology in mouse models. We injected brain-derived FTLD-TDP-GRN extracts, with proven seeding activity in vitro, into the brains of previously characterized Dox-regulatable transgenic (Tg) mice expressing a cytoplasmic NLS mutant of human TDP-43 (hTDP43$_{NLSm}$) in forebrain neurons (CamKIIa-hTDP43$_{NLSm}$)[25,26]. These CamKIIa-hTDP43$_{NLSm}$ mice develop little to no pathological TDP-43 aggregates in the brain even late in their lifespan[26]. Transgene expression was suppressed by providing Dox-containing chow,

and 1 week prior to brain extract injection, mice were switched to standard chow to induce transgene expression. Brain-derived FTLD-TDP-GRN extracts were stereotaxically injected into the neocortex, hippocampus, and thalamus and sacrificed 1 month post-injection (mpi). Brain sections were subjected to immuno-histochemistry (IHC) to detect pathological TDP-43 using the phospho-specific p409-410 antibody (Supplementary Fig. 5a, b and Supplementary Table 3).

We detected pTDP-43-positive neuronal cytoplasmic inclusions (NCIs) in the neocortex (Fig. 2a–d) and hippocampus (Fig. 2e–n) as early as 1 mpi, with more pathology ipsilateral rather than contralateral to the injection site. Cortical regions also exhibited both perikaryal and neuritic p409-410 immunoreactive pathology. Notably, pTDP-43-positive aggregates were more abundant in deep neocortical layers relative to superficial neocortical layers at 1 mpi (Fig. 2a, b).

The hippocampus also exhibited pTDP-43 aggregates in neuronal perikarya and their processes with significantly greater abundance ipsilateral compared with contralateral side (Fig. 2e–n). Pyramidal CA1 neurons exhibited pTDP-43 aggregates in the cytoplasm (Fig. 2e, f), in apical dendrites within striatum radiatum (Fig. 2e), and basal dendrites within stratum oriens (Fig. 2e). Dentate gyrus (DG) granule neurons also showed pTDP-43 cytoplasmic aggregates and within the distal mossy fiber (MF) processes (Fig. 2e, g). At this time point, caudal hippocampal areas exhibited fewer pTDP-43 NCIs relative to those closer to the injection. However, pTDP-43 staining was observed in the processes and cytoplasm of neurons in the subiculum (Sub), a more distant region from the injection sites but strongly connected with CA1 neurons (Fig. 2k, l). Notably, the contralateral hemisphere showed pTDP-43 inclusions in scattered neurons with relatively faint neuritic staining in cortical (Fig. 2c, d) and hippocampal neurons (Fig. 2h–j, m, n), indicating that TDP-43 pathology spreads to distal sites in the opposite hemisphere at 1 mpi.

Staining of adjacent sections with p409-410 (Fig. 2o, p) and another antibody that recognize the phospho-epitope Ser403/Ser404 (Fig. 2q, r) revealed that TDP-43 aggregates were phosphorylated at multiple sites. Furthermore, we examined the recruitment of hTDP-43$_{NLSm}$ (Fig. 2s, hTDP-43) vs. endogenous mouse TDP-43 proteins (Fig. 2t, mTDP-43) into NCIs by IF using species-specific antibodies. As previously described[26], hTDP-43$_{NLSm}$ protein is localized diffusely in neuronal cytoplasm and processes because of the NLS mutation (Fig. 2s). However, in injected mice, pTDP-43-positive aggregates recruited hTDP-43$_{NLSm}$ proteins into NCIs (Fig. 2s, left panels). As expected, endogenous nuclear mTDP-43 was decreased in cells expressing

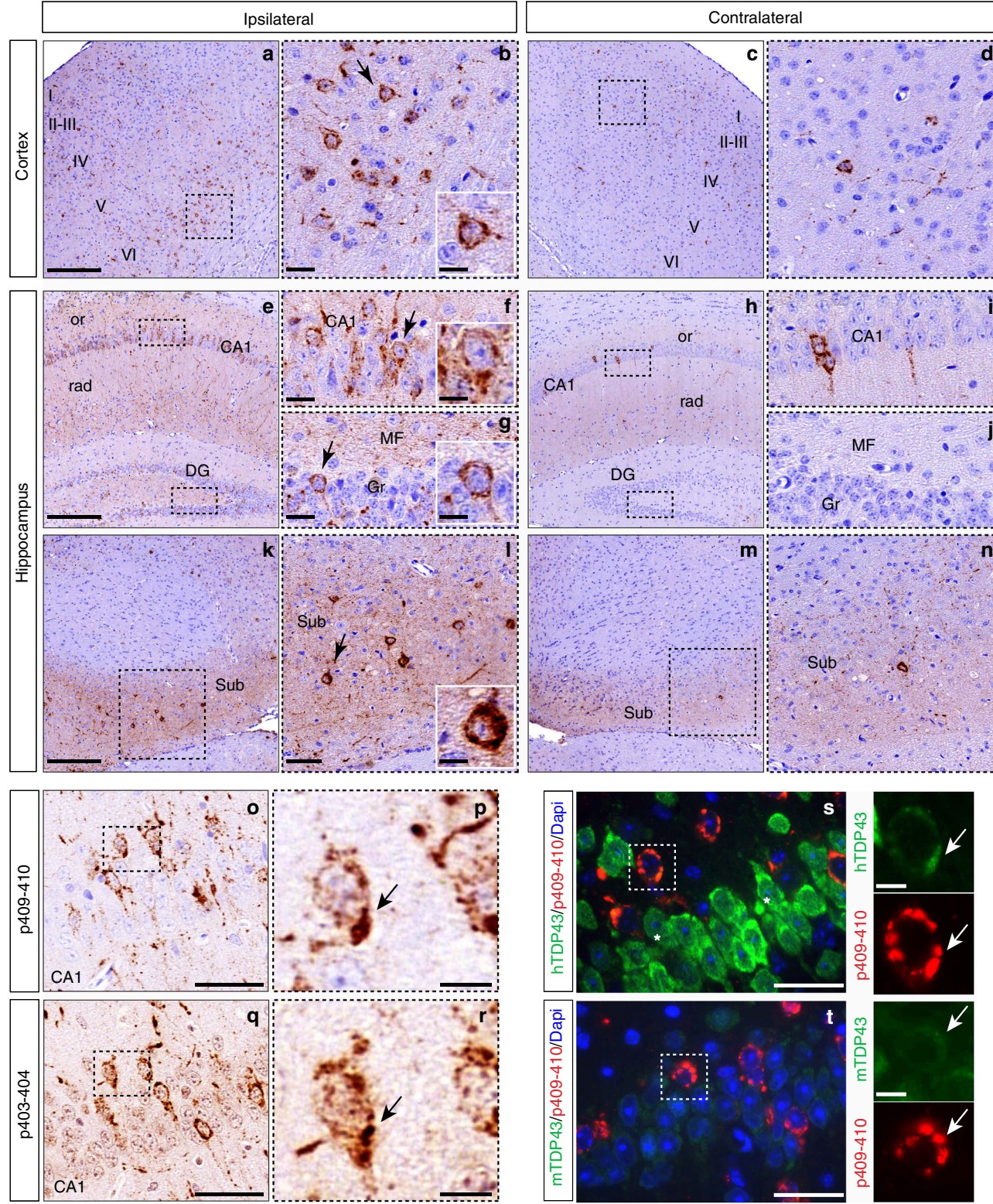

the hTDP-43$_{NLSm}$ transgene due to TDP-43 autoregulation (Fig. 2t)[26]. Thus no significant recruitment of mTDP-43 into pTDP-43-positive NCIs was detected (Fig. 2t, left panels).

The seeding of de novo TDP-43 pathology in CamKIIa-hTDP-43$_{NLSm}$ mice was validated by injecting brain-derived extracts from sporadic FTLD-TDP, FTLD-TDP-C9+ and additional FTLD-TDP-GRN cases (Supplementary Fig. 6a–c and Supplementary Table 3). Intracerebral injections of equivalent amounts of total insoluble protein from age-matched healthy brains did not induce TDP-43 pathology (Supplementary Fig. 6d), indicating that seeding of TDP-43 pathology was not a non-specific reaction against exogenous human protein extracts. Additionally, TDP-43 pathology was not significantly different between male and female animals.

We confirmed these in vivo data using another TDP-43 Dox-regulatable Tg mouse model expressing hTDP-43$_{NLSm}$ under the

**Fig. 2** FTLD-TDP-GRN extracts induce de novo TDP-43 pathology in CamKIIa-hTDP-43$_{NLSm}$ mice. Representative photomicrographs of p409-410 IHC staining in the cortex (layers I–VI, **a**–**d**) and hippocampus (**e**–**n**) of injected CamKIIa-hTDP-43$_{NLSm}$ mice at 1 mpi (n = 3) in ipsilateral and contralateral sides of the brain. Higher magnifications of the black-dashed boxes in **a**, **c** show p409-410-positive staining in cortical cells in **b**, **d**. Arrow in **b** points to the cell magnified in the inset. **e**–**n** show p409-410 immunostaining in the hippocampus: CA1 layer, stratum radiatum (rad), stratum oriens (or), dentate gyrus (DG), and subiculum (Sub). Higher magnifications of the black-dashed boxes in **e**, **h**, **k**, **m** show CA1 pyramidal cells (**f**, **i**), granule cells (Gr) and mossy fibers (MF) (**g**, **j**) and Sub (**l**, **n**). Arrows in **f**, **g**, **l** point to cells magnified in the insets. **o**–**r** show p409-410 and p403-404 IHC staining in adjacent sections of hippocampus from injected CamKIIa-hTDP-43$_{NLSm}$ mice at 1 mpi. **p**, **r** show higher magnifications of the black-dashed boxes in **o**, **q**, respectively, and arrow points to the same neuron bearing NCIs positive for p409-410 (**p**) and p403-404 (**r**). **s**, **t** show representative double-label IF images of p409-410 (red) staining and hTDP-43 (**s**, green) and mouse TDP-43 (**t**, green). Asterisks in **s** indicate the primary cytoplasmic distribution of hTDP-43$_{NLSm}$ protein. Panels at the far right depict higher magnifications of the white-dashed boxes in **s** and **t**, showing co-localization with aggregates positive for p409-410 (red, white arrow) and hTDP-43$_{NLSm}$ (green, white arrow) or nuclear clearance and reduced recruitment of mTDP-43 (green, white arrow) into p409-410-positive aggregates (red, white arrow). Scale bar = 200 μm (**a**, **c**, **e**, **h**, **k**, **m**), 50 μm (**l**, **m**, **o**, **g**, **s**, **t**) 20 μm (**b**, **d**, **f**, **g**, **i**, **j**) and 10 μm (right panels in **s**, **t** and insets)

control of the neurofilament heavy subunit (NEFH) promoter (rNLS8)[27]. Using the same dosing and timing schedule as described above (Supplementary Fig. 5a, c and Supplementary Table 3), rNLS8 mice recapitulated the cortical and hippocampal pTDP-43 pathology seen in CamKIIa-hTDP-43$_{NLSm}$ mice (Supplementary Fig. 7a–n). rNLS8 mice also exhibited a few scattered perikaryal pTDP-43 inclusions in the ipsilateral thalamus (Supplementary Fig. 7o–p) not found in CamKIIa-hTDP-43$_{NLSm}$ mice. No pathological TDP-43 seeding was observed in rNLS8 mice injected with equivalent amounts of insoluble protein from control brains or phosphate-buffered saline (PBS) alone (Supplementary Fig. 8).

**TDP-43 spreading pattern supports transneuronal transmission.** To further examine the spread of TDP-43 pathology, we analyzed over time the distribution of pTDP-43 in the CamKIIa-hTDP-43$_{NLSm}$ Tg mice injected with FTLD-TDP-GRN extracts (Supplementary Fig. 9b and Supplementary Table 3). At 1 mpi in the cortex, ipsilateral pTDP-43 NCIs were most abundant in deep neocortical layers, whereas aggregates progressively spread into superficial layers at 3, 5, and 9 mpi (Fig. 3a–d). Interestingly, the relatively granular staining of pTDP-43 in the cytoplasm and proximal neurites at early time points (Fig. 3a) progressed to become more compact at 3, 5, and 9 mpi (Fig. 3b–d). Moreover, there was an increase of pTDP-43 pathology in wider areas of the contralateral cortex over time (Fig. 3e–h), indicating the time-dependent spread of TDP-43 pathology to sites distant from the injection.

At 1 mpi, CA1 hippocampal neurons adjacent to the injection site showed relatively granular cytoplasmic and neuritic pTDP-43 staining, similar to that observed in neocortical neurons (Fig. 3i). Over time, this pathology became more condensed in neuronal perikarya with a remarkable reduction of pTDP-43 pathology in distal processes of the stratum radiatum and stratum oriens (Fig. 3j–l). The relatively granular staining in apical CA1 neurites at 1 mpi progressed to become thicker and fragmented (Fig. 3i–l), reflecting either the degeneration of these pTDP-43-positive processes or clearance of TDP-43 pathology over time. The contralateral hippocampus also exhibited progression of TDP-43 pathology with an increase in neuritic pTDP-43 immunoreactivity in stratum radiatum and oriens in addition to NCIs over time (Fig. 3m–p).

De novo formation of TDP-43 inclusions was observed in other neuronal populations over time. For example, MF projections from DG granule neurons exhibited TDP-43 pathology at 1 mpi in contrast with adjacent CA3 neurons of the hippocampus, which were unaffected (Fig. 3q). This transitioned over time to a loss of MF pathology at 3 mpi and the concomitant formation of relatively granular cytoplasmic

pTDP-43 staining in CA3 neurons (Fig. 3r) followed by the formation of more compact cytoplasmic inclusions at 5 and 9 mpi (Fig. 3s, t). A similar time-dependent increase in cytoplasmic pTDP-43 staining was observed in contralateral CA3 neurons (Fig. 3u–x). These results suggest that TDP-43 pathology spreads via the connectome of the injection site to different neuronal populations which then undergo a maturation process from diffuse granular deposits to condensed compact inclusions.

To further characterize TDP-43 pathology over time, we analyzed the presence of insoluble pTDP-43 protein in both ipsilateral and contralateral hippocampi at 1, 3, and 5 mpi by immunoblot (Supplementary Fig. 10a). At 1 mpi, more insoluble and phosphorylated full-length TDP-43 as well as three major CTFs was detected in the ipsilateral hippocampus (Supplementary Fig. 10a). At 3 and 5 mpi, insoluble pTDP-43 increased in the ipsilateral and contralateral hippocampus confirming the spreading of TDP-43 pathology to distal areas from the injection.

Double IF analysis using the N-terminal TDP-43 (N1065) and p409-410 antibodies confirmed the presence of p409-410-positive NCIs that occasionally co-localizes with full-length TDP-43 protein in the hippocampus of injected CamKIIa-hTDP-43$_{NLSm}$ mice (Supplementary Fig. 10b). Interestingly, many aggregates lack co-localization between N1065 and p409-410, suggesting that NCIs consisted primarily of phosphorylated CTFs (Supplementary Fig. 10b).

To document the distribution of de novo TDP-43 pathology over time in injected CamKIIa-hTDP-43$_{NLSm}$ mice, we constructed heat maps to visualize the spatial distribution. NCIs and neuritic pTDP-43 immunostaining in gray and white matter tracts were graded on a scale from 0 to 3 to represent absent to the most abundant pathology and color-coded in six rostral-to-caudal coronal sections at 1, 3, 5, and 9 mpi (Fig. 4). These heat maps reveal the spreading pattern of TDP-43 pathology from the ipsilateral side of the brain at earlier time points to greater involvement of wider brain areas in both ipsilateral and contralateral hemispheres over time (Fig. 4). In addition to illustrating pTDP-43 in cortical and hippocampal areas rostral and caudal to the injection, these heat maps also show TDP-43 accumulation in other more widespread brain regions in a time-dependent manner, such as the nucleus accumbens (Fig. 5a–d), the lateral septum (Fig. 5e–h), and others. Interestingly, TDP-43 pathology was not restricted to gray matter as it also increased in a time-dependent manner in axons of white matter tracts, including the dorsal fornix (Fig. 5i–l) and pyramidal tracts (Fig. 5m–p) among others.

To evaluate whether the pTDP-43 inclusions are found in glia as well as in neurons, double IF was performed to assess co-localization of TDP-43 pathology with markers specific for neurons (NeuN), astrocytes (glial fibrillary acidic protein (GFAP)), microglia (Iba1), and oligodendrocytes (Olig2)

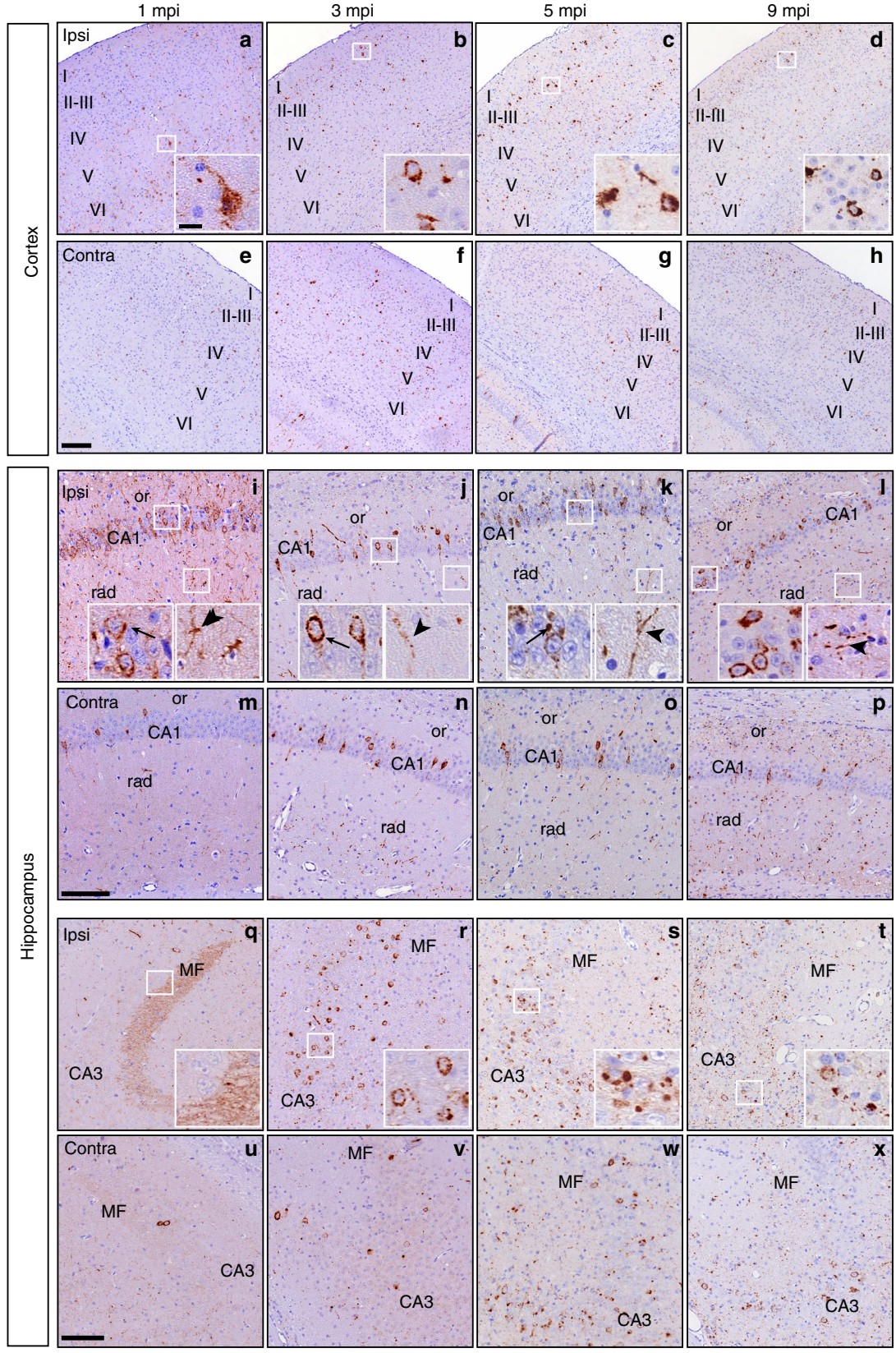

(Supplementary Fig. 11). In gray matter, double IF analysis confirmed that neurons accumulate the most abundant pTDP-43-positive inclusions (Supplementary Fig. 11a–c) with no appreciable co-localization of pTDP-43 with GFAP (Supplementary Fig. 11d–f) and Iba1 (Supplementary Fig. 11g–i). Only at later time points were rare pTDP-43 aggregates found to co-localize in the fimbria with oligodendroglial markers (Supplementary Fig. 11j–l) or in the dorsal hippocampal commissure with astrocytic markers (Supplementary Fig. 11m–o). Since the hTDP-43$_{NLSm}$ transgene is driven by the neuron-specific CamKIIa

**Fig. 3** TDP-43 pathology distribution in the cortex and hippocampus in CamKIIa-hTDP43$_{NLSm}$ mice. Representative photomicrographs of p409-410 IHC staining in coronal brain sections of cortex (layers I–VI, **a–h**) and hippocampus (**i–x**) from injected CamKIIa-hTDP43$_{NLSm}$ mice analyzed at 1 mpi ($n = 3$), 3 mpi ($n = 2$), 5 mpi ($n = 3$), and 9 mpi ($n = 3$) in the ipsilateral (Ipsi) and contralaleral (Contra) side of injection. In the cortex, **a–h** show p409-410 staining in cortical layers, insets are higher magnifications of the white boxes in **a–d**. In the hippocampus, **i–p** show p409-410-positive staining in CA1 layer, stratum radiatum (rad), and stratum oriens (or). Insets are higher magnifications of the white boxes in **i–l**. Arrows point to p409-410-positive NCIs in CA1 layer and arrowheads point to p409-410-positive neuronal neuritic processes in rad. **q–x** show p409-410 staining in CA3 layer and mossy fibers (MF). Insets are a higher magnifications of the white boxes in **q–t** showing p409-410-positive staining in MF (**q**) and CA3 neurons (**r–t**). Scale bar = 200 μm (**a–h**), 100 μm (**i–x**), and 20 μm (insets)

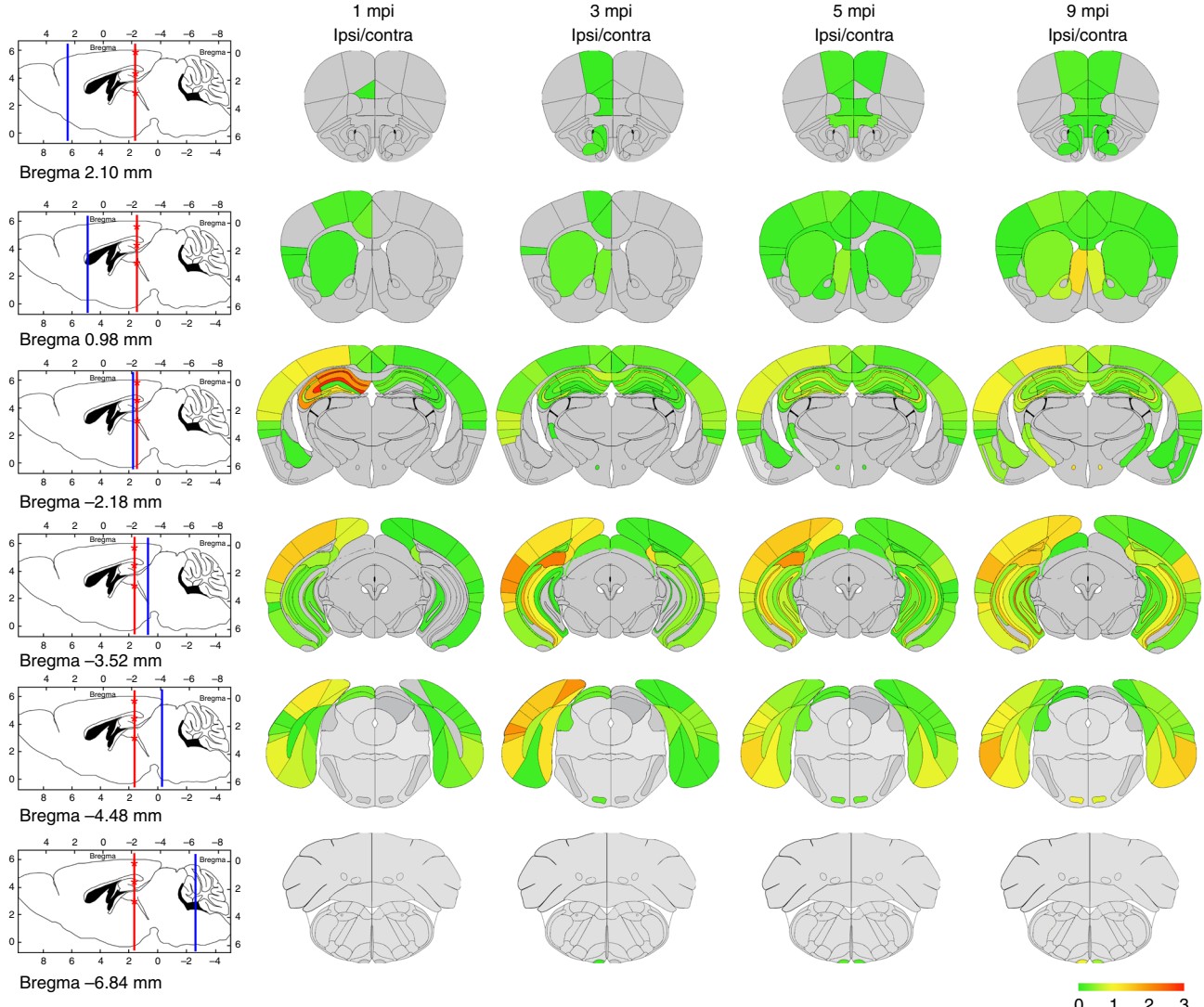

**Fig. 4** Heat maps show time-dependent spreading pattern of TDP-43 pathology. Heat maps show the distribution and burden of TDP-43 pathology in neurons and in white matter tracts in the brains of CamKIIa-hTDP-43$_{NLSm}$ mice at different post-injection times (1, 3, 5, and 9 mpi) in ipsilateral (Ipsi) and contralateral (Contra) sides. The panels in the far left column illustrate sagittal views of the corresponding coronal planes (blue line) and three sites of the injection into cortex, hippocampus, and thalamus (red asterisks). Each panel represents a coronal plane (Bregma; 2.10 mm, 0.98 mm, −2.18 mm, −3.52 mm, −4.48 mm, and −6.84 mm) for the different post-injection times. The burden of TDP-43 pathology was graded from negative (0, gray) to most abundant (3, red) and color-coded (scale bar) in the six representative coronal sections illustrated here depicting rostral to causal brain regions. Heat maps represent the mean values of the grades TDP-43 pathology from $n = 3$ mice at 1 mpi, $n = 2$ mice at 3 mpi, $n = 3$ mice at 5 mpi, and $n = 3$ mice at 9 mpi

promoter, it is not surprising to find predominantly neuronal TDP-43 pathology in this model.

**TDP-43 seeding in vivo recapitulates FTLD-TDP pathology.** TDP-43 inclusions in FTLD-TDP and ALS brains are ubiquitinated and immunoreactive for p62/SQTM1[7]. Thus we asked

whether the de novo TDP-43 aggregates in injected CamKIIa-hTDP-43$_{NLSm}$ mice recapitulate the neuropathological features of human TDP-43 aggregates. We show that pTDP-43 NCIs are also positive for p62 and ubiquitin at all time points, including the earliest time of 1 mpi (Supplementary Fig. 12). These results reveal that injection of brain-derived FTLD-TDP extracts

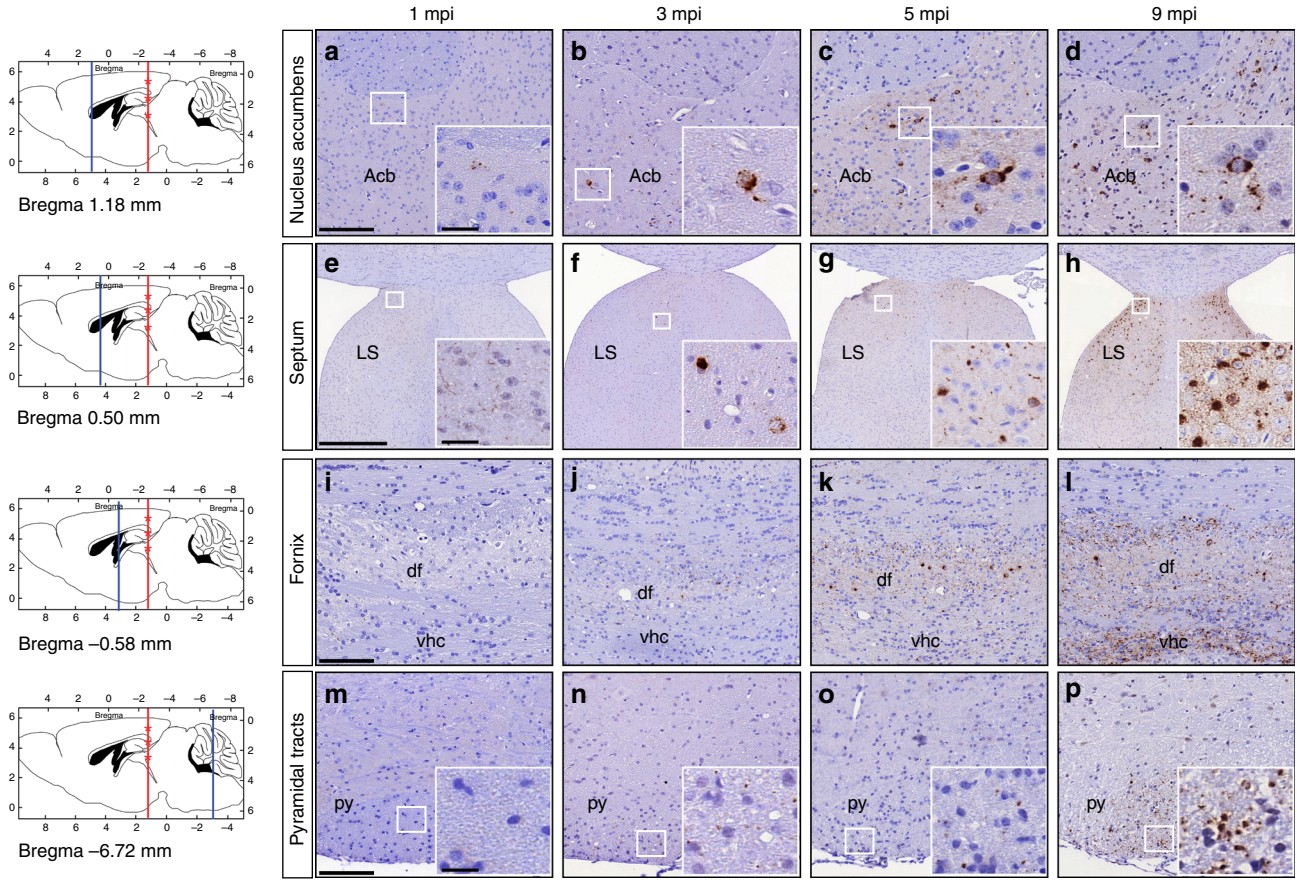

**Fig. 5** TDP-43 pathology spreading to rostral deep brain nuclei and white matter tracts. Representative photomicrographs of p409-410 IHC staining in rostral deep brain nuclei distal from the injection sites and white matter tracts over time in CamKIIa-hTDP-43<sub>NLSm</sub> mice at 1 mpi (*n* = 3), 3 mpi (*n* = 2), 5 mpi (*n* = 3), and 9 mpi (*n* = 3). The panels in the far left column illustrate sagittal views of the corresponding coronal planes (blue line) and three sites of the injection into cortex, hippocampus, and thalamus (red asterisks). p409-410 IHC staining in nucleus accumbens (Acb) (**a–d**, bregma 1.18 mm), septum (**e–h**, lateral septum (LS), bregma 0.50 mm), and white matter tracts in fornix (**i–l**, dorsal fimbria (df) and ventral hippocampal commissure (vhc), bregma −0.58 mm) and pyramidal fiber tracts (Py) (**m–p**, Bregma −6.72 mm). Insets are higher magnifications of the white boxes in **a–h** and **m–p**. Scale bar = 100 μm (**a–d**, **i–l**, and **m–p**), 500 μm (**e–h**) and 20 μm (insets)

activates proteostasis pathways in response to pathologic TDP-43, similar to what is seen in FTLD-TDP/ALS brains.

One plausible mechanism for the nucleation and seeding of TDP-43 pathology is the assembly of stress granules (SGs) that could trap RNA-binding proteins such as TDP-43 into the cytoplasm[28–35]. Double IF failed to show co-localization of the SG components G3BP1[36] and TIAR[37] with pTDP-43-positive NCIs in injected CamKIIa-hTDP-43<sub>NLSm</sub> mice (Supplementary Fig. 13).

**TDP-43 mislocalization enhances seeding and spreading in vivo**. In FTLD-TDP brains, CTFs are a major component of cytoplasmic TDP-43 pathology[38]. While the C-terminal region of TDP-43 is essential for aggregation and self-templating, the N-terminus of TDP-43 is also important in regulating its oligomerization and biological function[39–41].

To investigate the contribution of CTFs in the seeding process, FTLD-TDP-GRN extracts were injected into the brains of a previously characterized Dox-regulatable Tg model expressing TDP-43 208-414-CTF (CamKIIa-208)[42] (Supplementary Fig. 9a, c). The formation of de novo pTDP-43 pathology was compared to CamKIIa-hTDP-43<sub>NLSm</sub> mice injected with the same amount of pathological TDP-43 seeds at 1 mpi (Supplementary Table 3). The extent of seeding in the injected CamKIIa-208 mice (Fig. 6a, d) was significantly lower than in CamKIIa-hTDP-43<sub>NLSm</sub> mice

(Fig. 6b, e) with cytoplasmic pTDP-43 staining in scattered cells within deep neocortical layer VI neurons (Fig. 6a) and a few pyramidal neurons in CA1 (Fig. 6d). This contrasts with much higher number of pTDP-43-positive NCIs in both the cortex (Fig. 6b) and hippocampus (Fig. 6e) of the injected CamKIIa-hTDP-43<sub>NLSm</sub> mice. Nonetheless, the presence of NCIs in injected CamKIIa-208 mice were clearly evident, distinct from the diffuse perikaryal p409-410 pattern previously described in this Tg line (Fig. 6d)[42].

To characterize whether 208-CTFs were the principal components of the pTDP-43-positive NCIs, we used a polyclonal antibody that preferentially recognizes hTDP-43 (Supplementary Fig. 14a). Double IF showed pTDP-43 inclusions immunopositive for hTDP-43 indicating that transgene-derived 208-CTFs were recruited into the NCIs (Supplementary Fig. 14b–d). Additionally, staining for endogenous mTDP-43 using two antibodies that recognize specifically mTDP43 (Supplementary Fig. 14a and e–j) and an N-terminal antibody (Supplementary Fig. 14a and k–m) revealed that mTDP-43 was cleared from the nucleus and faintly co-localized with pTDP-43-positive NCIs (Supplementary Fig. 14e–m).

Finally, we evaluated whether FTLD-TDP-GRN brain lysates promote the recruitment of endogenous mTDP-43 in non-Tg WT mice (B6C3HF1) at 1 and 9 mpi (Supplementary Fig. 9a, d). We detected rare pTDP-43-positive NCIs in the

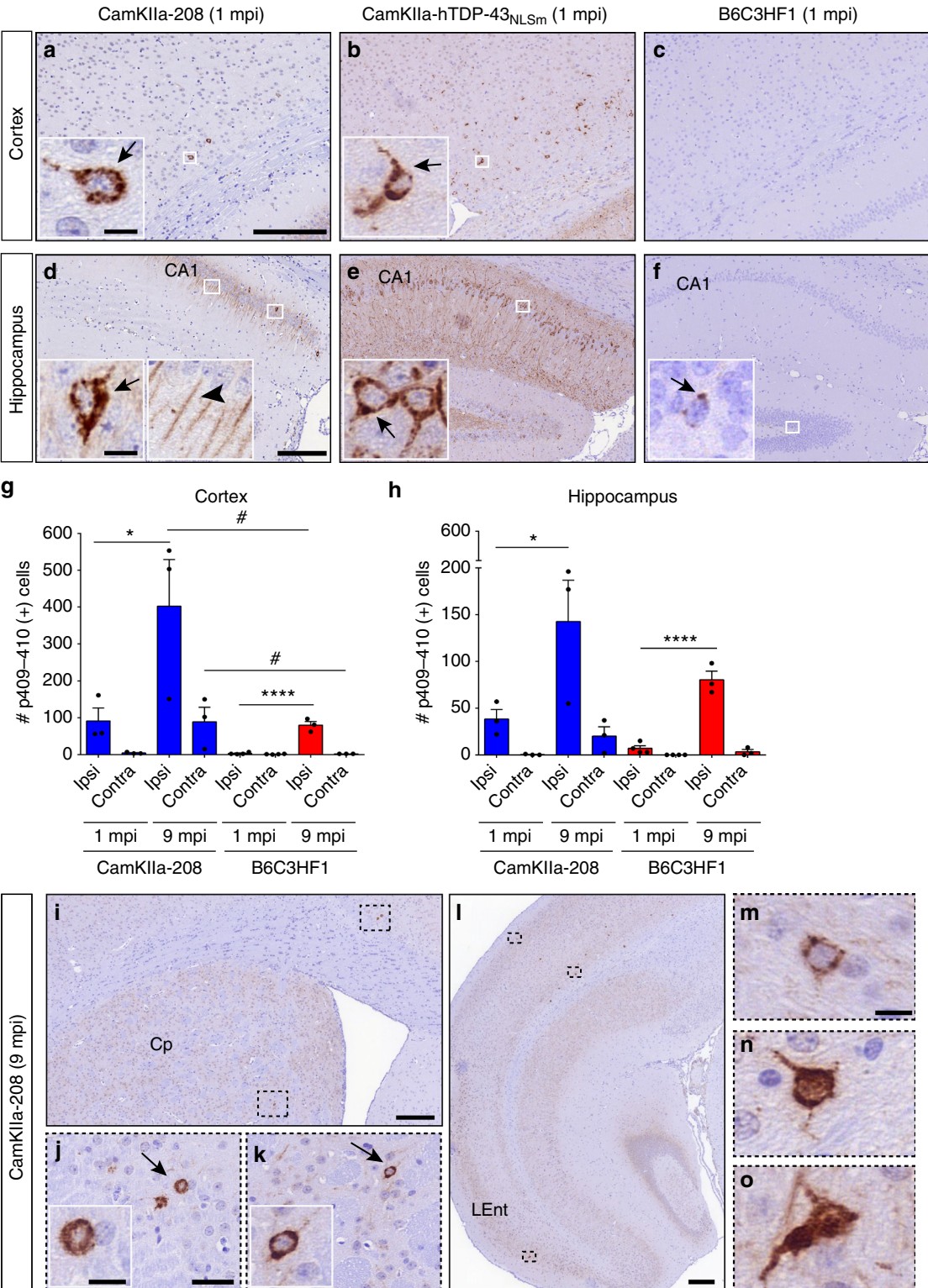

cortex or hippocampus of injected non-Tg mice at 1 mpi (Fig. 6c, f), with NCIs increasing over time by 9 mpi in the ipsilateral cortex and hippocampus. Quantification of pathology in injected non-Tg and CamKIIa-208 mice revealed a significant increase in the number of cells with pTDP-43 NCIs in the ipsilateral cortex and hippocampus with a trend toward an increase in the contralateral side in CamKIIa-208 mice (Fig. 6g, h). However, the dramatic increase in pTDP-43 NCIs in the cortex of CamKIIa-208 mice vs. non-Tg mice at 9 mpi

indicates that expression of 208-CTF facilitates the spreading of pathological TDP-43. Moreover, we observed the spreading of TDP-43 pathology in CamKIIa-208 mice to distal brain regions rostral and caudal to the injection site with pTDP-43 NCIs in the cingulate cortex (Fig. 6i, j) and caudate/putamen (Fig. 6i, k) at 9 mpi. Additionally, scattered pTDP-43-positive neurons were found in superficial and deep cortical layers (Fig. 6l–n) and in the lateral entorhinal cortex (Fig. 6l, o).

**Fig. 6** TDP-43 mislocalization increases seeding efficiency and spread of pathology. Representative photomicrographs of p409-410 IHC staining in the cortex and hippocampus of CamKIIa-208 (**a**, **d**), CamKIIa-hTDP-43$_{NLSm}$ (**b**, **e**) and B6C3HF1 (**c**, **f**) mice at 1 mpi. Insets are higher magnifications of white framed areas. Arrows and arrowhead points to p409-410-positive NCIs and dendrites, respectively. **g**, **h** Plot shows number of p409-410-positive cells in the cortex and hippocampus in Ipsi and Contra sides of CamKIIa-208 (blue bars) and B6C3HF1 mice (red bars) at 1 and 9 mpi. Bar plots show mean and whiskers s.e.m., with individual points representing different mouse overlaid as black circles. Two-way ANOVA followed by a Sidak's multiple comparisons test was used to compare time points (Factor, P value (F, degrees of freedom)); in **g** CamKIIa-208 (Cx): side $P = 0.019$ (8.55, 1), time $P = 0.020$ (8.32, 1), and interaction $P = 0.137$ (2.72, 1); B6C3HF1 (Cx): side $P = 0.008$ (11.97, 1), time $P = 0.028$ (7.12, 1), and interaction $P = 0.105$ (3.32, 1). In **h** CamKIIa-208 (Hp): side $P < 0.0001$ (85.98, 1), time $P < 0.0001$ (81.74, 1), and interaction $P < 0.0001$ (77.6, 1); B6C3HF1 (Hp): side $P < 0.0001$ (94.6, 1), time $P < 0.0001$ (78.8, 1), and interaction $P < 0.0001$ (65.69, 1). *$P < 0.05$, ****$P < 0.0001$. Two-way ANOVA followed by Sidak's multiple comparisons test was used to compare genotypes; in **g** Cx (ipsi); genotype $P = 0.007$ (11.71, 1), time $P = 0.010$ (10.42, 1), and interaction $P = 0.008$ (3.78, 1); Cx (contra): genotype $P = 0.033$ (6.26, 1), time, $P = 0.041$ (5.63, 1), and interaction $P = 0.0456$ (5.37, 1). In **h** Hp (Ipsi): genotype $P = 0.054$ (4.88, 1), time $P = 0.002$ (17.57, 1), and interaction $P = 0.483$ (0.53, 1); Hp (contra): genotype $P = 0.107$ (3.20, 1), time $P = 0.038$ (5.86, 1), and interaction $P = 0.119$ (2.95, 1). #$P < 0.05$. **i**–**o** show p409-410 immunostaining in areas distal from the injection site in CamKIIa-208 mice at 9 mpi. Higher magnifications of black framed areas in **i** and **l** show p409-410 NCIs in cingulate cortex (**j**), caudate putamen (Cp) (**k**), superficial and deep cortical layers (**m**, **n**, respectively), and lateral entorhinal cortex (Lent) (**o**). Arrows in **j** and **k** point to p409-410-positive neurons magnified in insets. Scale bar = 200 μm (**i**, **l**), 50 μm (**j**, **k**) and 10 μm (**m**–**o** and insets)

Overall our data indicate that mislocalization of cytoplasmic TDP-43 protein, particularly full-length TDP-43, is a key factor for an efficient seeding and spreading of TDP-43 pathology in vivo.

## Discussion

Neuropathological studies describing the stereotyped, progressive regional involvement of TDP-43 pathology in FTLD-TDP, ALS, AD, and other neurodegenerative disorders support the idea that TDP-43 aggregates may be transmitted from cell to cell resulting in the spread of neuropathological inclusions affecting interconnected brain regions over time[4,43]. Although pathological TDP-43 from FTLD-TDP and ALS brains can act as seeds to promote the formation and propagation of TDP-43 aggregates in cultured cells[15,16], transmission and spreading of TDP-43 pathology using animal models has not been reported. Here we show that injections of human brain-derived FTLD-TDP extracts containing pathological TDP-43 into the brains of TDP-43 Tg mice expressing mislocalized cytoplasmic TDP-43 induces de novo TDP-43 pathology. Remarkably, this TDP-43 pathology spreads within the brain in a spatio-temporal manner consistent with a model of cell-to-cell transmission of pathology that appears to occur transneuronally through the neuroanatomical connectome.

These in vivo observations were enabled by experiments in which a stable cell line expressing cytoplasmic TDP-43 mutant GFP-NLSm was used to screen brain-derived FTLD-TDP extracts that actively seeded TDP-43 for in vivo studies. Initially, two different Dox-inducible cell lines with tightly regulated expression of either nuclear GFP-WT or GFP-NLSm proteins were generated. Consistent with other cell culture studies[15,16,24], we found that overexpression of TDP-43 facilitates the seeding of TDP-43 aggregates using pathological TDP-43 from FTLD-TDP brain extracts. Moreover, we show that the seeding and aggregation of TDP-43 protein is more efficient when TDP-43 is mislocalized in the cytoplasm. Nonaka et al.[16] showed seeding and aggregation of TDP-43 in SH5Y5 cells transiently transfected with WT or NLS mutant TDP-43[16], although differences in the recruitment of these exogenously expressed proteins was not described. This discrepancy could be due to differences in transgene expression levels between the two in vitro systems. Although TDP-43 is predominantly nuclear, it has been shown that transient transfection increases cytoplasmic TDP-43[19], which could increase the seeding of WT TDP-43. In agreement with the previous studies, our data show that both the concentration and the subcellular localization of TDP-43 protein are important for TDP-43 aggregation and supports the hypothesis that disturbances in

nucleo-cytoplasmic TDP-43 homeostasis may play a role in the nucleation and aggregation of TDP-43 protein.

The screening for seeding activity of brain-derived pathological TDP-43 showed that FTLD-TDP-GRN lysates have the highest activity followed by FTLD-TDP-C9+ and sporadic FTLD-TDP cases. FTLD-TDP cases can be classified into at least five subtypes (types A–E) with a close correlation between pathologic subtype and genetic risk factors or clinical phenotypes[44–46]. Nonaka et al.[16] previously described in a cell-based system that brain-derived pathological TDP-43 from type A and B cases seeded TDP-43 protein into aggregates more effectively than type C[16]. However, our cohort of sporadic and familial FTLD-TDP cases were classified as type A or B[44,45], and no specific subtypes explained the differences found in in vitro seeding activity. Nonetheless, we cannot rule out that *GRN* haplo-insufficiency may provide a unique milieu to promote the formation of pathological TDP-43 species with distinct seeding properties. Notably, strain-like properties of misfolded proteins have been well described for pathological α-syn[47,48], Aβ[49,50], and tau[51] and recent studies have shown evidence for the existence of strains in TDP-43 proteinopathies[17,24].

Biochemical analysis of pathological TDP-43 from human FTLD-TDP brain extracts revealed a correlation between reduced seeding activity and lack of two lower Mr CTFs[20,52]. Moreover, the resistance and banding pattern of CTFs after PK digestion suggested that distinct conformational TDP-43 species recovered from FTLD-TDP brains may be more potent. Indeed, synthetic peptides corresponding to specific sequences in the extreme C-terminus of TDP-43 form more active amyloid-like fibrils seeding TDP-43[17]. However, more extensive biochemical and biophysical studies are needed to demonstrate the presence of distinct pathogenic TDP-43 strains in FTLD-TDP brains due to assembly of different C-terminal TDP-43 regions. Such studies would strengthen the hypothesis that clinical heterogeneity in FTLD-TDP patients is due to the heterogeneity of TDP-43 neuropathology resulting from different TDP-43 strains.

Remarkably, our study demonstrates that brain-derived FTLD-TDP extracts not only efficiently induce seeding of de novo TDP-43 pathology when injected into the brains of TDP-43 Tg mouse models expressing cytoplasmic hTDP-43 protein but also to a lesser extent in WT mice. More importantly, our data demonstrate that TDP-43 pathology spreads throughout the brain in a spatio-temporal manner. The propagation of TDP-43 pathology in injected CamKIIa-hTDP-43$_{NLSm}$ mice resulted in widespread cortical, hippocampal, and subcortical pTDP-43 accumulation in both ipsilateral and contralateral hemispheres in a time-dependent manner, consistent with cell-to-cell transmission of TDP-43 pathology which follows the neuroanatomical

connectome from the injection site. For example, soon after injections of pathological brain-derived TDP-43 lysates into the anteromedial visual cortex, pTDP-43-positive NCIs were found within deep neocortical neurons within layer VI of the lateral visual cortex and lateral and dorsal auditory cortex. Anterograde tracer studies have demonstrated that pyramidal cells in ante-romedial visual cortex indeed project their axons to the lateral visual cortex and the lateral and dorsal auditory cortex via deep cortical neurons within layer VI (Allen Mouse Brain Connectivity Atlas and ref. [53]), consistent with the hypothesis that pathological TDP-43 species could propagate via axonal connections[4,5].

The pattern of TDP-43 pathology in the hippocampus also supports the hypothesis of axonal spreading of pathology. The absence of pTDP-43 in the CA3 layer at early time points fol-lowed by progressively increased numbers of TDP-43 aggregates in CA3 neurons over time supports the hypothesis of cell-to-cell transmission of pathology either by anterograde transport from DG granule cells via MF or retrograde transport from CA1 pyr-amidal neurons. These observations can be extended to include the spread of pTDP-43 pathology over time in dorsal hippo-campus, pre- and para-subiculum, and lateral and medial entorhinal cortex. Thus the spatio-temporal pattern of TDP-43 pathology spread in the hippocampus and related structures appears to be related to their neuronal connectivity[54].

While the spreading of TDP-43 pathology in our animal models is partly dependent on the distribution of transgene expression, TDP-43 pathology was found to progressively affect subcortical areas such as nucleus accumbens, the basolateral amygdala, and the caudate/putamen, areas also affected in FTLD-TDP brains[4,55,56]. This suggests that select neuronal populations may be particularly vulnerable to TDP-43 aggregates. Moreover, we also found that white matter oligodendrocytes and astrocytes are capable of developing pTDP-43 aggregates over time. This is particularly interesting since the CamKIIa promoter should direct transgene expression only in neurons. How TDP-43 pathology spreads to glial cells is not clear but could be due to neuron-to-glial transfer of TDP-43 pathology. Glial cells do accumulate TDP-43 inclusions in FTLD and ALS brains[4,13,57], while astro-cytes accumulate TDP-43 pathology in Alexander's disease[58]. Further studies are required to better understand the relationship between neuronal and glial TDP-43 and how it relates to cell-to-cell transmission of TDP-43 and other neurodegenerative disease pathologies.

Notably, as has been described in vitro[39–41], our data indicate that N-terminal sequences of TDP-43 may play an important role in self-templating and aggregating TDP-43 in vivo. Injections of pathological TDP-43 into the brains of CamKIIa-208 mice that overexpress cytoplasmic 208-CTFs did not efficiently seed and transmit TDP-43 pathology compared to CamKIIa-hTDP-43$_{NLSm}$ mice. While the expression levels of cytoplasmic 208-CTF and hTDP-43$_{NLSm}$ proteins are comparable, the high insolubility of 208-CTF protein[42] when compared with hTDP-43$_{NLSm}$[26] may also determine the accessibility of the pathological TDP-43 seeds, thereby affecting the efficiency of seeding and propagation of TDP-43 pathology in CamKIIa-208 mice.

Our study also demonstrates the seeding of endogenous TDP-43 and propagation of TDP-43 pathology over time in non-Tg mice injected with brain-derived FTLD-TDP-GRN seeds, albeit at a very low rate. Our findings suggest that pathophysiological conditions, such as aging or cellular stress affecting homeostasis and accumulation of cytoplasmic TDP-43, may promote the seeding and propagation of TDP-43 pathology in the brain. Although disease-causing missense mutations promoting cyto-plasmic mislocalization of TDP-43 are uncommon in ALS cases[59], defects in the nucleocytoplasmic function have been proposed as a common pathogenic mechanism in ALS and

FTLD[60]. C9orf72 animal models demonstrate impaired nucleo-cytoplasmic transport[61–63] and a link between cytoplasmic poly-GA dipeptide, a major aggregating protein in C9orf72 patients, and impairment of TDP-43 nuclear import has been reported[64]. Furthermore, TDP-43 aggregates in cultured cells causes the disruption of nuclear membrane and nuclear pore complexes, reducing both nuclear protein import and mRNA export[65]. Overall these results suggest that TDP-43 aggregates may result in a positive feedback loop due to nucleocytoplasmic transport defects.

Additional studies are required to better elucidate the mechanisms that affect the cell-to-cell transmission of misfolded TDP-43 as well as the processes that affect TDP-43 homeostasis. The mouse models we have developed here provide compelling systems for the interrogation of molecular mechanisms and physiological consequences of transneuronal spreading of TDP-43 disease pathology. This raises the intriguing idea that inhi-biting the cell-to-cell transmission of TDP-43 pathology may mitigate disease progression and that blocking transmission of TDP-43 pathology may have therapeutic potential for the treat-ment of TDP-43 proteinopathies.

## Methods

**Constructs**. N-terminal GFP-tagged TDP-43: GFP-WT and GFP-NLSm, were generated by subcloning the cDNA encoding WT TDP-43 and site-directed mutagenized NLS sequences of TDP-43 (NLS1/2, K82A/R83A/K84A-K95A/K97A/R98A)[18] into the enhanced GFP-C1 vector (Clontech, Mountain View, CA). cDNA for GFP-WT and GFP-NLSm were subcloned into the tetracycline (Tet)-inducible expression vector, pLVX-Tight-Puro, using the BamHI and NotI restriction sites to generate the pLVX-GFP-WT- and pLVX-GFP-NLSm-expressing vectors.

**Generation of Dox-inducible iGFP-WT and iGFP-NLSm stable cell lines**. iGFP-WT and iGFP-NLSm cells were generated by transient transfection of pLVX-GFP-WT and pLVX-GFP-NLSm expression vectors into QBI-293 stable cells expressing the reverse tetracycline transcriptional activator rtTA (QBI-293rtTA cells (clone #6.10)) generated in the Center for Neurodegenerative Disease Research (CNDR). QBI-293A cells were originally purchased from Quantum (AES0506). QBI-293rtTA cells were maintained in Dulbecco's modified Eagle's medium (Corning Cellgro, Manassas, VA) supplemented with 10% of tetracycline-screened fetal bovine serum (FBS; Atlanta Biologicals, Flowery Branch, GA), 1% Penicillin–Streptomycin (Corning Cellgro, Manassas, VA), and L-glutamine (20 mM, Corning Cellgro, Manassas, VA) with G418 (400 µg ml−1, Calbiochem, La Jolla, CA). Cells were maintained at 37 °C in humidified atmosphere containing 5% $CO_2$. For transfection, cells were plated at $3 \times 10^5$ cells in 6-multi-well plates the day before and medium was replaced with 500 µl of pre-warmed growth medium immediate before transfection. Transfection mixture (500 µl) contained 15 µl of FuGENE® 6 Transfection Reagent (Roche Applied Science, Indianapolis, IN) and 3.0 µg of either pLVX-GFP-WT or pLVX-GFP-NLSm vectors in Opti-MEM® (Thermo Scientific Inc., Rockford, IL). 175 µl of Opti-MEM® was removed from each well and was replaced with 175 µl of transfection mixture. After 4 h incuba-tion, transfection medium was replaced with growth medium. Cells were selected with puromycin (2 µg ml−1, Sigma, St Louis, MO) and drug-resistant cells were subcloned by limited dilution in 96-well plates. iGFP-WT (clone #12.5) and iGFP-NLSm (clone #6.B7) individual subclones were expanded and transgene expression was validated by IF and immunoblot.

**Autopsy cohort**. Human postmortem brains were obtained from the University of Pennsylvania, CNDR Brain Bank[66]. Histopathologic subtyping of our FTLD-TDP cohort was conducted according to established guidelines[44,45] and genetic testing for GRN mutations and C9orf72 expansions was performed[67,68]. All necessary written informed consent forms were obtained from the patients or their next of kin in accordance with University of Pennsylvania Institutional Review Board guidelines and confirmed at the time of death. The genetic and demographic data of cases used in this study are summarized as Supplementary Table 1.

**Mouse cohorts**. Monogenic tetO-hTDP-43$_{NLSm}$ and tetO-208 were bred with CamKIIa-tTA mice or NEFH-tTA to generate bigenic mice. Genotyping from tail DNA was performed using the following set of primers: CamKIIa-rtTA-frw (CGCTGTGGGGCATTTTACTTTAG) and CamKIIa-rtTA-rev (CATGTCCAGA TCGAAATCGTC); NEFH-rTA-frw (ATTGAGGGCTTGACCACCAGGAA) and NEFH-rTA-rev (GCATGTCCAGGTCAAAATCATCCAAG) and for hTDP-43$_{NLSm}$ and 208 transgenes; TDP-43-frw (TTGGTAATAGCAGAGGGGGTGG AG) and MoPrP-rev (TCCCCCAGCCTAGACCACGAGAAT)[26,27,42].

Mice were maintained with chow containing Dox (200 mg kg$^{-1}$, Dox Diet, S3888; Bio-Serv., Flemington, NJ). Transgene expression was induced by switching to standard chow lacking Dox (Rodent Diet 20 #5053, PicoLab, St. Louis, MO). WT B6C3HF1 mice were purchased from the Jackson Laboratories (Bar Harbor, ME). Both male and female adult mice (3–5 months old) were used for this study. All breeding, housing, and procedures were performed according to the NIH Guide for the Care and Use of Experimental Animals and approved by the University of Pennsylvania Institutional Animal Care and Use Committee (IACUC).

**In vitro TDP-43 aggregation assay**. For the analysis of the efficiency of seeding of iGFP-WT and iGFP-NLSm cells, the same amounts of sarkosyl-insoluble extracts were used for transduction; if not specified in the text, 1.0 µg total protein per well (24-MW plates) or 4.0 µg total protein per well (6-MW plates) were used for transduction (Supplementary Table 2). Briefly, sarkosyl-insoluble extracts were extensively sonicated (30 pulses, intensity 2.5) and 8.0 µg protein was diluted to 100 µl with PBS (dPBS) and mixed with single-use tubes of Bioporter as a protein delivery reagent following the manufacturer's instructions (Bio-PORTER$^{TM}$, AMS Biotechnology, Milton, UK). Immediately before transduction, cells were washed twice with Opti-MEM® medium, which had been pre-equilibrated in the cell culture incubator overnight. Protein–Bioporter complexes were added to the cells and incubated for 4 h. Cells were placed back on fresh medium in the absence or presence of Dox (Dox +, 1.0 µg ml$^{-1}$) and cultured for 3 additional days (3 dpt). To compare the seeding activities of sporadic FTLD-TDP, FTLD-TDP-GRN, and FTDL-TDP-C9+ brain extracts, equivalent amounts of insoluble TDP-43 (250 pg) according to ELISA (C89) measures (Supplementary Table 2) were transduced into iGFP-NLSm cells. In order to transduce the same ratio of TDP-43 protein:total protein (0.01% µg TDP-43 µg$^{-1}$ total protein), sarkosyl-insoluble extracts from a non-pathological case were used to account for differences in protein content (2.5 µg total protein per well). Equivalent amounts of sarkosyl-insoluble extracts from non-pathological cases were used as a negative control (2.5 µg total protein per well). All cell culture experiments were blinded with respect to the patient-derived brain extracts used for transduction and posterior analysis.

**Sequential biochemical fractionation of cell extracts**. To evaluate the expression levels of induced GFP-WT and GFP-NLSm fusion proteins, iGFP-WT or iGFP-NLSm cells were plated in 6-MW plates (1.5 × 10$^5$ cells), and after 24 h, growth medium was replaced with pre-warmed growth medium with Dox (Dox+) or without Dox (Dox−) for an additional 72 h. Cells were washed with ice-cold dPBS and extracted with 300 µl of ice-cold RIPA buffer (50 mM Tris pH 8.0, 150 mM NaCl, 1% NP-40, 5 mM EDTA, 0.5% sodium deoxycholate, 0.1% sodium dodecyl sulfate (SDS)) containing 1 mM phenylmethylsulfonyl fluoride (PMSF) and a mixture of protease inhibitors and phosphatase inhibitors by sonication using a hand-held probe (13 pulses, 2.5 intensity) and centrifuged at 100,000 × g for 30 min at 4 °C. The supernatants were saved for analysis as RIPA fraction (R) and pellets were re-sonicated in 60 µl of Urea buffer (7 M urea, 2 M thiourea, 4% CHAPS, and 30 mM Tris, pH 8.5). After centrifugation at 100,000 × g for 30 min at 22 °C, the supernatant was saved as the Urea fraction (U). For evaluating the formation of insoluble TDP-43 aggregates after transduction, iGFP-WT and iGFP-NLSm cells were washed with ice-cold dPBS and protein was extracted with 300 µl homogenization buffer (HB: 10 mM Tris-HCl, pH 7.5 containing 0.8 M NaCl, 1 mM EGTA, 1 mM dithiothreitol) and 1 mM PMSF and protease inhibitor cocktail. Sarkosyl was added to the lysates to a final concentration of 1% and incubated for 10 min at room temperature. Protein extracts were sonicated and centrifuged at 100,000 × g for 30 min at 4 °C. The supernatant containing the sarkosyl-soluble protein was saved, whereas pellet was washed with 300 µl of dPBS, sonicated, and centrifuged again at 100,000 × g for 30 min at 4 °C. The final pellet was resuspended in 50 µl of dPBS as sarkosyl-insoluble fraction. Protein concentrations of the RIPA fractions were determined using the BCA (Thermo Scientific Inc., Rockford, IL).

**Immunoblot analyses**. To evaluate the expression levels of induced GFP-WT and GFP-NLSm fusion proteins, 20 µg of RIPA-soluble protein and a volume equivalent to 5.0 times the amount of the corresponding urea-soluble fraction was loaded on 10% SDS-polyacrylamide gel electrophoresis gels. The biochemical analyses of TDP-43 in the sarkosyl-insoluble fraction from human brain or iGFP-WT and iGFP-NLSm transduced cells and in the RIPA-insoluble fraction from mouse brain tissue were performed on 12% Bis–Tris gels (NuPAGE Novex, Thermo Scientific Inc., Rockford, IL) using MOPS-SDS as a running buffer. Proteins were transferred onto 0.45 µM nitrocellulose membranes and blocked with Odyssey blocking buffer (LI-COR Biotechnology, Lincoln, NE). Membranes were immunoblotted overnight at 4 °C with primary antibodies diluted in Odyssey blocking buffer (Supplementary Table 4). After three washes in 0.1 M Tris buffered saline with 0.1% Tween-20 (0.1 M Tris-0.1%Tween), membranes were further incubated with the corresponding pair of secondary antibodies (Supplementary Table 4). Membranes were then washed three times in 0.1 M Tris–0.1% Tween and finally scanned using an ODY-2816 Imager. The optical densities were measured with the 483 Image Studio software (LI-COR Biotechnology, Lincoln, NE).

**Stereotaxic surgery on mice**. Adult mice (3–5 months old) were deeply anesthetized with ketamine hydrochloride (100 mg kg$^{-1}$), xylazine (10 mg kg$^{-1}$), and acepromazine (0.1 mg kg$^{-1}$) and immobilized in a stereotaxic frame (David Kopf Instruments). Human sarkosyl-insoluble extracts were sonicated (60 pulses, intensity 2.5) and aseptically delivered unilaterally in the thalamus (rate of 0.6 µl min$^{-1}$), dorsal hippocampus (rate of 0.6 µl min$^{-1}$), and overlying cortex (rate of 0.4 µl min$^{-1}$) (Bregma: −2.5 mm; lateral: +2 mm; depth: −3.4 mm, −2.4 mm and −1.4 mm, respectively from the skull)[69] (Supplementary Figs. 5a and 9a) using a Hamilton® syringe (Hamilton, NV). Each injection site received 2.5 µl of sarkosyl-insoluble human extracts (Supplementary Table 3) or PBS.

**Mouse tissue collection**. Mice were lethally anesthetized using ketamine/xylazine/acepromazine and intracardially perfused first with PBS and then perfused-fixed with 10% neutral buffered formalin (NBF) for 20 min. The brains and spinal cord were dissected and immersion-fixed in NBF overnight at 4 °C. Post-fixed tissues were rinsed in Leaching buffer (50 mM Tris, 150 mM NaCl, pH 8.0) and brains were cut in 2-mm-thick coronal slices using a mouse brain slice matrix (Harvard Apparatus, Holliston, MA). Brain and spinal cord slices were processed for paraffin embedding.

**IF and IHC**. Distribution of GFP-WT and GFP-NLSm proteins and the presence of p409-410-positive TDP-43 aggregates (pTDP-43) in iGFP-WT and iGFP-NLSm stable cells were evaluated by IF. Cells were plated on coated poly-D-lysine coverslips and transduced as described above. Cells were washed once with dPBS at room temperature and fixed with 4% paraformaldehyde (4% PFA) for 15 min and washed three times with dPBS at room temperature. For experiments where the percentage of area occupied by p409-410 immunostaining was quantified, cells were fixed in 4% PFA containing 1% Triton X-100 to remove soluble proteins and to visualize the insoluble TDP-43 aggregates. After incubation in blocking buffer (3% bovine serum albumin–3% FBS in dPBS) for 30 min at room temperature, cells were incubated with specific primary antibodies (Supplementary Table 4) overnight at 4 °C in a humidified chamber. After washing with dPBS, cells were incubated with appropriate secondary antibodies in blocking buffer for 1 h at room temperature in the dark. Coverslips were mounted with Fluoromount G$^{TM}$ containing DAPI (4,6-diamidino-2-phenylindole; Thermo Scientific Inc., Rockford, IL). Images were obtained in a high-resolution Leica DMI6000B microscope using the Leica LAS-X software.

For IHC studies, paraffin-embedded 6-µm-thick sections were deparaffinized in xylene and rehydrated in graded alcohol concentrations[27,70]. Endogenous peroxidases were quenched by incubating sections in 5% H$_2$O$_2$/methanol for 30 min at room temperature. After washing in water for 10 min, antigen retrieval was performed in 1% Antigen Unmasking Solution (Vector) by microwaving for 15 min at 99 °C. Slides were then washed with 0.1 M Tris buffer, pH 7.6 for 5 min. To reduce nonspecific signals, brain sections were blocked with 2% FBS–0.1 M Tris buffer and immunostained with the corresponding primary antibody (Supplementary Table 4) overnight at 4 °C in a humidified chamber. Sections were incubated with a biotin-conjugated secondary antibody diluted in blocking buffer for 1 h at room temperature (Supplementary Table 4). Antigen–antibody reactions were visualized using VECTASTAIN AB solution and ImmPACT DAB solution (Vector Laboratories Inc., Burlingame, CA). Hematoxylin-counterstained slides were dehydrated through graded alcohol concentrations and xylene and mounted with Cytoseal (Thermo Scientific Inc., Rockford, IL)[27,70]. Bright-field images were acquired at ×20 magnification using a Lamina Multilabel slide scanner (Perkin Elmer, Waltham, MA). For double-labeled IF, deparaffinized sections were blocked as described above and incubated with different combinations of primary antibodies overnight at 4 °C (Supplementary Table 4). Secondary antibodies were diluted in blocking buffer (Supplementary Table 4) and sections were incubated for 2 h at room temperature in a humidified chamber. To reduce endogenous autofluorescence, sections were immersed in a 0.3% Sudan black–70% ethanol solution for 15–20 s, followed by a wash in water for 10 min. Finally, sections were mounted in Vectashield-hard medium containing DAPI (Vector Laboratories Inc., Burlingame, CA). Images were obtained in a high-resolution Leica DMI6000B microscope using the Leica LAS-X software.

**Heat maps**. The semiquantitative heat maps for injected CamKIIa-hTDP-43$_{NLSm}$ mice were generated using stained slides with p409-410 antibody. Briefly, pTDP-43 pathology was graded from negative (0, gray) to more abundant (3, red) and color-coded into six representative coronal sections from rostral to caudal brain levels (Bregma; 2.10, 0.98, −2.18, −3.52, −4.48. and −6.84 mm). Brain regions were marked according to the anatomical patterns in The Mouse Brain Atlas in stereotaxic coordinates[69]. After grading individual brain regions for each mouse, averaged values for each time point analyzed (1 mpi (n = 3), 3 mpi (n = 2), 5 mpi (n = 3), and 9 mpi (n = 3)) were used to generate heat maps using the in-house software to graphically show distribution of pTDP-43 pathology based on a scale-color system[71].

**Quantification of TDP-43 pathology**. To quantify the extent of TDP-43 aggregation in iGFP-WT or iGFP-NLSm cells transduced with brain-derived human extracts, iGFP-WT or iGFP-NLSm stable cells were plated into coverslips and fixed 3 dpt with 1% Tx–4% PFA to remove soluble proteins and immunostained using the p409-410 antibody. Whole coverslips were scanned and images were digitalized

using a Lamina Multilabel Slide Scanner (Perkin Elmer, Waltham, MA). The images were processed using the Indica Labs HALO software and the percentage of area occupied by p409-410 immunopositive staining[72] was quantified and used as a measure of the seeding activity in our cell-based system. The number of DAPI-positive nuclei (#Dapi $\mu m^{-2}$) was quantified using the Indica Labs HALO software.

To evaluate the extent and spread of TDP-43 pathology in CamKIIa-208 and WT (B6C3HF1) mice injected with FTLD-TDP-GRN extracts, bright-field images of brain sections immunostained with p409-410 antibody were acquired at ×20 magnification and digitized using a Lamina Multilabel slide scanner (Perkin Elmer, Waltham, MA). Cells containing cytoplasmic p409-410-positive staining were manually counted in the cortex and hippocampus using the CaseViewer software (3DHistech). For each animal, p409-410-positive cells were counted in a total of 50–60 coronal brain sections separated by 120 μm and covering rostral to caudal areas of the brain.

**Statistics**. For each experiment, the corresponding statistics test is indicated in the figure legend. Number of samples for each group or how many times the experiment was repeated independently is indicated in the figure legend. Paired and unpaired two-tailed Student's $t$ test was performed when two groups were compared and one-way analysis of variance (ANOVA) or two-way ANOVA was performed when multiple groups were compared, according to different conditions. Statistical analysis was performed using the Prism software 6.0 (GraphPad Software, La Jolla, CA).

## Data availability

All relevant data are available from the corresponding author upon reasonable request. Uncropped immunoblot images are available as Supplementary Figure 15.

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

## Acknowledgements

The authors specially thank the patients and their families for their contributions. We thank T. Schuck, J. Robinson, and C. Casalnova for assistance with obtaining suitable brains for these studies; Dr. V. Van Deerlin and Dr. E. Suh for the genetic analysis; C. Li, T. Khan, M. Dominique, M. Olufemi, and R. J. Gathagan for technical assistance; T. Lehr and M. Rawal for helping with tissue sectioning; M. D. Byrne and A. Yousef for assistance with image digitalization; Dr. K. Brunden, Dr. B. Pettmann, Dr. P. Weinreb, Dr. J. Coomaraswamy, and Dr. D. Irwin for helpful discussions; Dr. Sharon Xie for advice on statistical analysis; Dr. P. Davies for providing the PHF1 antibody; and Dr. M. Neumann and Dr. E. Kremmer for providing the p409-410 antibody (TAR5P-1D3). This work was supported by NIH grants AG10124 and AG17586, the Wyncote Foundation, and a grant from Biogen.

## Author contributions

S.P. designed and performed the experiments, analyzed the data, and wrote the manuscript. Y.X., C.R.R. and L.K.W. performed experiments and analyzed the data; B.Z. and H.J.B. performed stereotaxic injections; E.B.L. and J.Q.T. critically reviewed and edited the manuscript. V.M.-Y.L. designed research, supervised the study, and wrote the manuscript.

## Additional information

**Competing interests:** The authors declare no competing interests.

