## [Peer Review File · Nature Communications]

REVIEWERS' COMMENTS:

Reviewer #1 (Remarks to the Author):

The authors have addressed some of my concerns raised earlier during the review process for Nature Medicine. I agree that some of my suggested mechanistic experiments may take substantial efforts to complete and the current manuscript is suited for publication in Nature Communication.

Reviewer #2 (Remarks to the Author):

This manuscript is improved from its initial form. The essential findings that TDP-43 can be seeded in vitro and in vivo by pathological forms of the TDP-43 protein are reasonably strong. Their failure to immunodeplete TDP-43 prior to injection in vivo makes the findings slightly less robust: within an animal it is not clear that the "toxic agent" is in fact aggregated TDP-43. However, given the effectiveness of immunodepletion in their studies in vitro, the conclusion that TDP-43 can induce pathology is reasonable. The paper as it stands will add to a substantial literature indicating that it is possible to induce protein pathology in vitro and in vivo through inoculation of aggregated forms. The manuscript is now suitable for publication.

Note: subject/verb mis-match in the abstract.

Reviewer #3 (Remarks to the Author):

The authors have addressed my major concerns, primarily in regards to the difference in PGRN extracts and weaknesses of the in vitro experiments. The manuscript is now appropriate for publication without further changes.